# PAI-1 Inhibitor TM5441 Attenuates Emphysema and Airway Inflammation in a Murine Model of Chronic Obstructive Pulmonary Disease

**DOI:** 10.3390/ijms26157086

**Published:** 2025-07-23

**Authors:** Kyohei Oishi, Hideki Yasui, Yusuke Inoue, Hironao Hozumi, Yuzo Suzuki, Masato Karayama, Kazuki Furuhashi, Noriyuki Enomoto, Tomoyuki Fujisawa, Takahiro Horinouchi, Takayuki Iwaki, Yuko Suzuki, Toshio Miyata, Naoki Inui, Takafumi Suda

**Affiliations:** 1Second Division, Department of Internal Medicine, Hamamatsu University School of Medicine, 1-20-1 Handayama, Chuo-ku, Hamamatsu 431-3192, Japan; kyoishi@hama-med.ac.jp (K.O.); yinoue@hama-med.ac.jp (Y.I.); hozumi@hama-med.ac.jp (H.H.); yuzosuzu@hama-med.ac.jp (Y.S.); karayama@hama-med.ac.jp (M.K.); k.furu@hama-med.ac.jp (K.F.); norieno@hama-med.ac.jp (N.E.); fujisawa@hama-med.ac.jp (T.F.); suda@hama-med.ac.jp (T.S.); 2Center for Clinical Research, Hamamatsu University Hospital, Hamamatsu University School of Medicine, 1-20-1 Handayama, Chuo-ku, Hamamatsu 431-3192, Japan; inui@hama-med.ac.jp; 3Department of Pharmaceutical Sciences, School of Pharmacy at Narita, International University of Health and Welfare, 4-3, Kozunomori, Narita 286-8686, Japan; horinouchi-takahiro-ea@iuhw.ac.jp; 4Department of Pharmacology, Hamamatsu University School of Medicine, 1-20-1 Handayama, Chuo-ku, Hamamatsu 431-3192, Japan; tiwaki@hama-med.ac.jp; 5Department of Medical Physiology, Hamamatsu University School of Medicine, 1-20-1 Handayama, Chuo-ku, Hamamatsu 431-3192, Japan; seigan@hama-med.ac.jp; 6Department of Molecular Medicine and Therapy, Tohoku University School of Medicine, 2-1 Seiryo-Machi, Aoba-ku, Sendai 980-8575, Japan; toshio.miyata.c8@tohoku.ac.jp; 7Department of Clinical Pharmacology and Therapeutics, Hamamatsu University School of Medicine, 1-20-1 Handayama, Chuo-ku, Hamamatsu 431-3192, Japan

**Keywords:** chronic obstructive pulmonary disease, cigarette smoke extract, neutrophil elastase, plasminogen activator inhibitor-1, plasminogen activator inhibitor-1 inhibitor, TM5441

## Abstract

Chronic obstructive pulmonary disease (COPD) is a major cause of morbidity and mortality worldwide, primarily driven by chronic airway inflammation due to cigarette smoke exposure. Despite its burden, however, current anti-inflammatory therapies offer limited efficacy in preventing disease progression. Plasminogen activator inhibitor-1 (PAI-1), as a key regulator of fibrinolysis, has recently been implicated in structural airway changes and persistent inflammation in patients with COPD. This study aimed to investigate the ability of the PAI-1 inhibitor TM5441 to attenuate airway inflammation and structural lung damage induced by a cigarette smoke extract (CSE) in a mouse model. Mice received intratracheal CSE or vehicle on days 1, 8, and 15, and were sacrificed on day 22. TM5441 (20 mg/kg) was administered orally from days 1 to 22. The CSE significantly increased the mean linear intercept, destructive index, airway resistance, and reductions in dynamic compliance. The CSE also increased the numbers of neutrophils and macrophages in the bronchoalveolar lavage fluid, systemic PAI-1 activity, and neutrophil elastase mRNA and protein expression in the lungs. TM5441 treatment significantly suppressed these changes without affecting coagulation time. These findings suggest that TM5441 may be a novel therapeutic agent for COPD by targeting PAI-1-mediated airway inflammation and emphysema.

## 1. Introduction

Chronic obstructive pulmonary disease (COPD) is a leading cause of morbidity and mortality worldwide and is currently ranked among the top three causes of death globally [1]. COPD affects approximately 10% of individuals over the age of 30 years [2], and its global burden is expected to increase further as a result of ongoing exposure to risk factors, particularly cigarette smoke, and a rapidly aging population [3]. Cigarette smoking remains the primary risk factor for COPD, contributing to epithelial injury [4] and promoting neutrophil recruitment to the lungs [5]. The hallmark pathological features of COPD include the accumulation of macrophages in the peripheral airways and lung parenchyma, accompanied by increased infiltration of neutrophils. Neutrophils release neutrophil elastase (NE), a major elastolytic enzyme that contributes to a protease–antiprotease imbalance [6], leading to small-airway lesions and emphysema, characterized by alveolar wall destruction [5,7]. NE also promotes the activation and recruitment of macrophages in the lungs [8]. Together with epithelial and other structural cells, these inflammatory cells release a wide array of inflammatory mediators, thereby contributing to the characteristic pathogenic features of COPD, including chronic inflammation and tissue destruction [9]. Inhaled long-acting bronchodilators remain the cornerstone of COPD management and can alleviate symptoms and reduce exacerbations, but fail to target the underlying inflammatory processes, and current pharmacological therapies have therefore shown limited success in altering the disease’s natural course or reducing mortality. There is thus an urgent need for novel therapies that target the fundamental mechanisms of COPD pathogenesis.

The serine protease inhibitor plasminogen activator inhibitor-1 (PAI-1) is the main physiological inhibitor of tissue-type and urokinase-type plasminogen activators. PAI-1 is present at low concentrations under normal physiological conditions, but its expression is markedly increased in various pathological states, including infection, myocardial infarction, and diabetes. These conditions are common comorbidities of COPD and are known to affect its clinical course. In addition to its antifibrinolytic role, PAI-1 has emerged as a profibrotic factor implicated in the pathogenesis of various pulmonary diseases [10], and elevated systemic levels of PAI-1 have been reported in patients with COPD, associated with accelerated lung function decline [11]. Tiwari et al. reported that PAI-1 played a key role in cigarette smoke-induced pulmonary inflammation, and that its genetic deficiency protected against this inflammatory response [12]. In addition, PAI-1 deficiency attenuated age-related airspace enlargement in a senescence mouse model [13]. Collectively, these findings suggest that PAI-1 may represent a promising therapeutic target in patients with COPD.

Several promising PAI-1 inhibitors have been identified. Among these, the orally available small-molecule inhibitor TM5275 has demonstrated efficacy in preclinical models of respiratory diseases, including bronchial asthma [14] and pulmonary fibrosis [15]. TM5441, as a derivative of TM5275, was designed to bind at the flexible joint region in PAI-1 and has demonstrated potent inhibitory activity in vivo [16]. The efficacy of PAI-1 inhibitors, however, has not yet been evaluated in models of cigarette smoke-induced COPD.

This study aimed to investigate the therapeutic potential of TM5441 in a murine model of COPD induced by intratracheal administration of cigarette smoke extract (CSE). We determined if TM5441 could attenuate the pathological features of COPD, including inflammation, pulmonary function decline, and emphysematous changes in lung tissue.

## 2. Results

### 2.1. Effects of Intratracheal Administration of CSE on Emphysema

We confirmed characteristic pathological features of COPD following intratracheal administration of CSE (Figure 1A). Representative histological images of lung sections stained with hematoxylin and eosin (HE) following intratracheal administration of phosphate-buffered saline (PBS) or CSE are shown (Figure 1B). Compared with the PBS group, mice in the CSE group exhibited a significantly higher mean linear intercept (MLI) (32.6 ± 1.4 μm vs. 44.7 ± 1.0 μm, *p* < 0.001; *n* = 5 per group) and destructive index (DI) (13.2 ± 1.4% vs. 44.1 ± 1.2%, *p* < 0.001; *n* = 5 per group), indicating airspace enlargement and alveolar wall destruction. These findings demonstrate that intratracheal administration of CSE induced pulmonary emphysema in mice (Figure 1B,C).

### 2.2. CSE-Induced Lung Functional Impairment

We also assessed the effects of intratracheal CSE on lung function. Compared with the PBS group, airway resistance (RI) was significantly increased (0.90 ± 0.01 cmH_2_O/mL/s vs. 1.04 ± 0.04 cmH_2_O/mL/s, *p* = 0.004; *n* = 8–10 per group) and dynamic compliance (Cdyn) was markedly decreased (0.027 ± 0.000 mL/cmH_2_O vs. 0.024 ± 0.001 mL/cmH_2_O, *p* = 0.023; *n* = 8–10 per group) in mice in the CSE group, indicating impaired pulmonary mechanics (Figure 1D). These findings are consistent with those reported in previous COPD models [17,18].

### 2.3. Effects of CSE on Total Cell, Macrophage, and Neutrophil Counts in the Bronchoalveolar Lavage Fluid (BALF)

We analyzed the BALF to assess airway inflammation, and representative flow cytometry plots are shown in Appendix A. Total cell (77 × 10^3^ ± 8 × 10^3^ vs. 173 × 10^3^ ± 20 × 10^3^, *p* < 0.001), macrophage (62 × 10^3^ ± 6 × 10^3^ vs. 117 × 10^3^ ± 5 × 10^3^, *p* < 0.001), and neutrophil counts (0.4 × 10^3^ ± 0.2 × 10^3^ vs. 13.3 × 10^3^ ± 5.4 × 10^3^, *p* = 0.022) in the BALF showed a significant increase in the CSE group compared with the PBS group (*n* = 7–8 per group) (Figure 1E). These results confirmed an increase in inflammatory cells in the BALF in CSE-treated mice.

### 2.4. Effects of CSE on PAI-1 Expression in Lung Tissues and PAI-1 Activity in Plasma

We evaluated the effects of CSE exposure on PAI-1 expression and activity by measuring PAI-1 mRNA levels in lung tissues by reverse transcription-polymerase chain reaction (RT-PCR) and plasma PAI-1 activity by enzyme-linked immunosorbent assay (ELISA). PAI-1 expression in lung tissues was significantly upregulated in the CSE group (*p* = 0.016; *n* = 5 per group). In addition, plasma PAI-1 activity was markedly elevated in the CSE group compared with the PBS group (0.40 ± 0.11 ng/mL vs. 0.96 ± 0.07 ng/mL, *p* = 0.003; *n* = 5 per group) (Figure 2).

### 2.5. Effects of TM5441 on CSE-Induced Pulmonary Emphysema

Given that PAI-1 activity was elevated in the COPD mouse model, we determined if the PAI-1 inhibitor TM5441 attenuated CSE-induced emphysema. TM5441 administration had no notable effect on the MLI in the control (PBS) group, but significantly decreased the MLI in the CSE group (55.8 ± 2.1 μm vs. 37.2 ± 1.4 μm, *p* < 0.001; *n* = 5 per group). Similarly, TM5441 had no effect on the DI in the PBS group, but a statistically significant decrease in the DI was observed in the CSE group (41.0 ± 1.7% vs. 23.4 ± 1.4%, *p* < 0.001; *n* = 5 per group). These results suggest that TM5441 suppressed CSE-induced airspace enlargement and alveolar destruction (Figure 3A–C).

### 2.6. Effects of TM5441 on CSE-Induced Respiratory Dysfunction

The effects of TM5441 on pulmonary function in the PBS and CSE groups are shown in Figure 3D. TM5441 did not affect RI or Cdyn in the PBS group, but significantly reduced the elevated RI (1.05 ± 0.08 cmH_2_O/mL/s vs. 0.90 ± 0.02 cmH_2_O/mL/s, *p* = 0.025; *n* = 5–8 per group) and markedly improved the reduced Cdyn (0.020 ± 0.002 mL/cmH_2_O vs. 0.027 ± 0.001 mL/cmH_2_O, *p* = 0.007; *n* = 5–8 per group) in the CSE group. These findings indicate that TM5441 suppressed CSE-induced respiratory dysfunction.

### 2.7. Effects of TM5441 on CSE-Induced Inflammatory Cell Infiltration in BALF

TM5441 treatment had no significant effect on the total cell, macrophage, or neutrophil counts in the BALF in the PBS group, but TM5441 significantly reduced the numbers of total cells (178 × 10^3^ ± 13 × 10^3^ vs. 117 × 10^3^ ± 11 × 10^3^, *p* = 0.001), macrophages (152 × 10^3^ ± 15 × 10^3^ vs. 109 × 10^3^ ± 11 × 10^3^, *p* = 0.029), and neutrophils (0.8 × 10^3^ ± 0.2 × 10^3^ vs. 0.2 × 10^3^ ± 0.1 × 10^3^, *p* = 0.015) in the CSE group (*n* = 5 per group) (Figure 4A). These results indicate that TM5441 attenuated CSE-induced airway inflammation in this model.

### 2.8. Effects of TM5441 on CSE-Induced PAI-1 Expression in Lung Tissue and Plasma PAI-1 Activity

We determined the effects of TM5441 on CSE-induced PAI-1 expression in lung tissue and plasma PAI-1 activity. TM5441 treatment did not alter PAI-1 mRNA expression in lung tissue in the CSE group, but did suppress the CSE-induced increase in plasma PAI-1 activity (1.73 ± 0.18 ng/mL vs. 1.23 ± 0.06 ng/mL, *p* = 0.042; *n* = 5 per group) (Figure 4B). These findings suggest that TM5441 did not affect the upstream regulation of PAI-1 expression, but directly inhibited PAI-1 activity.

### 2.9. Effects of TM5441 on CSE-Induced NE mRNA Expression and Protein Levels in Lung Tissue

We evaluated NE expression in lung tissue at the mRNA and protein levels. TM5441 significantly decreased NE mRNA expression in lung tissue lysates in the CSE group (*p* = 0.008; *n* = 5–7 per group), determined by quantitative RT-PCR, and NE protein levels (4305 ± 1236 pg/mL of lung homogenate vs. 1626 ± 212 pg/mL of lung homogenate, *p* = 0.011; *n* = 5–7 per group), assessed by ELISA (Figure 5). These findings suggest that TM5441 attenuated neutrophil inflammation in CSE-induced lung injury by downregulating NE.

### 2.10. Effects of TM5441 on Coagulation Parameters

We evaluated the effects of CSE on coagulation parameters to assess its safety. The prothrombin time and activated partial thromboplastin time were similar in TM5441-treated and untreated mice (8.46 ± 0.08 s vs. 8.50 ± 0.10 s, *p* = 0.995; 28.62 ± 0.81 s vs. 28.22 ± 1.20 s, *p* = 0.995, respectively; *n* = 4–5 per group) (Figure 6). Plasma fibrinogen levels were also comparable between the two groups (193.6 ± 13.3 mg/dL vs. 188.2 ± 13.9 mg/dL, *p* = 0.990). These findings suggest that TM5441 did not adversely affect coagulation function, supporting its favorable safety profile.

## 3. Discussion

This study demonstrated that intratracheal administration of CSE induced the key pathological and physiological features of COPD in mice, including morphological alterations in the lung, respiratory dysfunction, and airway inflammation characterized by macrophage and neutrophil infiltration. Notably, both systemic PAI-1 activity and its expression in lung tissue were elevated following CSE exposure. Treatment with the novel PAI-1 inhibitor TM5441 significantly ameliorated these pathological changes by reducing systemic PAI-1 activity and suppressing NE expression in lung tissue at both the mRNA and protein levels, together with improved physiological factors. These findings suggest a mechanistic link between PAI-1 activity and cigarette smoke-induced pulmonary inflammation and emphysema. Importantly, TM5441 attenuated CSE-induced pathophysiological changes without prolonging coagulation times, highlighting its potential safety as a therapeutic agent.

To reliably evaluate the therapeutic potential of TM5441, it is essential to utilize a COPD model that faithfully reproduces the pathological and functional features of the disease. COPD is a progressive respiratory disorder associated with substantial global morbidity and mortality, for which cigarette smoking is the main risk factor. The disease is characterized by chronic inflammation, predominantly involving macrophages and neutrophils, and is typically resistant to corticosteroid therapy [19]. Current COPD treatment thus relies mainly on inhaled long-acting bronchodilators. Several anti-inflammatory drugs have been considered as candidates for COPD treatment. However, the efficacy of broad anti-inflammatory approaches remains limited, with only modest reductions in COPD exacerbations. High-dose inhaled corticosteroids carry an increased risk of pneumonia [20,21], and phosphodiesterase-4 inhibitors are associated with gastrointestinal side effects [22]. The oral neutrophil elastase inhibitor AZD9668 showed no clinical benefit in patients with COPD [23], and the development of neutrophil elastase inhibitors has not progressed further. Despite the urgent need for novel agents that specifically target COPD-associated inflammation, progress has been limited, partly because of an incomplete understanding of the mechanisms underlying disease onset and progression. Animal models that accurately recapitulate the structural and functional characteristics of COPD are therefore essential for therapeutic evaluation. Emphysema and small-airway lesions are hallmark pathological features of COPD. Emphysema is defined by alveolar wall destruction and enlargement of the alveolar spaces, which can be assessed quantitatively in animal models using morphological indices such as MLI and DI [24]. In our model, both MLI and DI were significantly increased following intratracheal CSE administration. In addition to emphysema, airway lesions are also key pathological features of COPD. Although pulmonary function tests are crucial for evaluating airway involvement, many murine models fail to demonstrate consistent functional impairments [25,26]; however, the current model showed increased RI and decreased Cdyn [17,18], indicating the presence of airway lesions. Furthermore, inflammatory cell infiltration, particularly by macrophages and neutrophils, plays a pivotal role in COPD-associated inflammation, and the numbers of these inflammatory cells in the BALF were significantly elevated in our model after CSE exposure. Collectively, these findings indicate that the current COPD model successfully recapitulates the key structural and functional features of the disease and is thus suitable for evaluating potential therapeutic interventions.

PAI-1 is a key regulator of the fibrinolytic system, exerting its effects within the vasculature and also across various organs. Its expression is markedly increased during both local and systemic inflammation. PAI-1 is associated with neutrophil activation [27] and inhibition of neutrophil efferocytosis [28], and acts as a chemotactic factor promoting neutrophil migration to sites of inflammation [29], thereby amplifying the inflammatory response. Recent studies have implicated PAI-1 in the pathogenesis of COPD. Systemic PAI-1 levels were elevated in patients with COPD [30], and this molecule is thought to contribute to airway inflammation and small airway remodeling, both of which are hallmark features of the disease. The major sources of PAI-1 in the lung are alveolar macrophages and epithelial cells [31]. Cigarette smoke, as the main risk factor for COPD, has been shown to cause alveolar epithelial injury and upregulate PAI-1 expression, thereby exacerbating pulmonary inflammation [12]. Immunohistochemical analyses have demonstrated PAI-1 expression in alveolar macrophages, with a positive correlation with collagen deposition in lung tissue from patients with COPD [32]. Elevated PAI-1 levels have also been detected in sputum samples from patients with COPD [31], and higher serum concentrations of PAI-1 have been linked to lung function decline and small-airway obstruction [11]. In line with these clinical findings, the current CSE-induced COPD model exhibited elevated systemic PAI-1 activity and increased PAI-1 mRNA expression in lung tissue. These results further support the involvement of PAI-1 in COPD pathophysiology and highlight its potential as a therapeutic target. Furthermore, PAI-1 is associated with common comorbidities of COPD, such as atherosclerosis and metabolic syndrome, and PAI-1 inhibition is considered a promising therapeutic approach in clinical practice.

The development of PAI-1 inhibitors has progressed, and examination of the X-ray crystal structure of the human PAI-1 molecule and a library of approximately 2 million virtual compounds has led to the identification of hit compounds, including TM5001 and TM5007, using an in silico approach [33]. Structural optimization led to the development of the lead compound TM5275 [34], which enhanced fibrinolysis on endothelial cells and effectively dissolved fibrin clots [35]. TM5275 also inhibited ovalbumin-induced neutrophil infiltration into the airways [14] and suppressed macrophage migration to sites of injury in animal models [36]. Further structural optimization led to the development of TM5441, with superior pharmacokinetic properties and a broader volume of distribution. TM5441 has demonstrated therapeutic efficacy in various animal models [37], including attenuating N^ω^-nitro-arginine methyl ester (L-NAME)-induced organ dysfunction, including alveolar enlargement [38]. However, the L-NAME model primarily reflects pulmonary changes resulting from long-term nitric oxide inhibition, rather than cigarette smoke-induced COPD. In contrast, the current CSE-induced COPD model more closely mimics the pathological changes observed in human cigarette smoke-related COPD, thereby providing a more clinically relevant context in which to evaluate the effects of TM5441. The current results demonstrated that PAI-1 inhibition with TM5441 suppressed CSE-induced morphological alterations, pulmonary dysfunction, and inflammatory cell infiltration in the BALF, without affecting coagulation times. TM5441 treatment did not alter PAI-1 mRNA expression in lung tissue following CSE administration, but it did suppress the CSE-induced increase in plasma PAI-1 activity (Figure 4B). These findings suggest that TM5441 does not affect the upstream regulation of PAI-1 expression but rather directly inhibits PAI-1 activity. Therefore, the possibility of an off-target effect of TM5441 appears unlikely. Collectively, these findings further support the therapeutic potential of PAI-1 inhibitors in patients with COPD, particularly for targeting cigarette smoke-induced airway pathology.

NE is a key inflammatory protease released by neutrophils and is strongly associated with the pathogenesis of COPD [39]. In this study, TM5441 reduced NE expression at both the mRNA and protein levels in lung tissues. A previous report demonstrated that PAI-1 knockout suppressed cigarette smoke-induced myeloperoxidase activity and the expression of C-X-C chemokines (CXCL1 and CXCL2) in lung tissues [12]. Furthermore, in a model of LPS-induced acute lung injury, increased neutrophil apoptosis was observed in the lungs of PAI-1 knockout mice [40]. These findings suggest that the mechanism by which TM5441 reduces NE mRNA and protein levels in the lungs is not only a reduction in neutrophil infiltration, but also inhibition of neutrophil activation and promotion of neutrophil apoptosis.

This study has several strengths. To our knowledge, this is the first report to demonstrate the efficacy of a PAI-1 inhibitor in a cigarette smoke-induced COPD model. The efficacy of TM5441 was comprehensively evaluated using histological, functional, and molecular analyses. Targeting PAI-1 represents a novel anti-inflammatory approach for COPD. In this study, TM5441 prevented the pathophysiological condition of COPD, and PAI-1 inhibition may represent a promising alternative therapeutic strategy that targets an upstream driver of both inflammation and tissue destruction. Furthermore, the inclusion of safety data is another strength point in this study. However, this study has some limitations. First, we used only male mice because sex differences in blood coagulation have been reported in mice [41,42]. Given that the prevalence of COPD is higher in males [43,44], male mice were selected to better reflect the clinical population. In addition, because this study involves a drug targeting the coagulation and fibrinolysis system, we standardized the sex of the animals used. Second, although we evaluated coagulation parameters as part of the safety analysis, bleeding time was not assessed. A previous report demonstrated that administration of TM5441 at doses of 50 and 100 mg/kg did not affect bleeding time in mice [45]. Therefore, we did not assess tail bleeding time in this study. Third, in our study, neutrophils were defined as CD11b^+^ Ly-6G/Ly-6C (Gr-1)^+^ cells using the RB6-8C5 antibody. While RB6-8C5 primarily binds Ly6G, it also cross-reacts with Ly6C. Thus, although the majority of this population is considered neutrophils, a minor fraction of Ly6C^hi^ monocytes may have been included. Fourth, this experimental model does not fully reflect COPD associated with chronic cigarette exposure. Furthermore, the timing of TM5441 administration limits the ability to evaluate its therapeutic effects. In future studies, we plan to use an experimental design in which drug administration begins after disease onset, in order to assess whether PAI-1 inhibition can reverse or halt the progression of existing COPD pathology. Finally, given that PAI-1 has profibrotic effects [46] and is associated with airway fibrosis, and that small-airway remodeling is a key pathological feature of COPD [47], it would be preferable to validate these findings using a model that more accurately reflects the pathophysiology of human COPD in order to better evaluate the therapeutic potential of TM5441. We plan to investigate whether TM5441 can prevent long-term small-airway remodeling and fibrosis, as well as evaluate its efficacy in female mice in future studies.

## 4. Materials and Methods

### 4.1. Animals and Experimental Protocol

Male C57BL/6NCr mice (7–9 weeks old) were purchased from Japan SLC (Japan SLC, Shizuoka, Japan) and housed in our facility at Hamamatsu University School of Medicine. All procedures were conducted in accordance with protocols approved by the Institutional Animal Care and Use Committee (Approval No. 22-009-03). We determined the group sizes with reference to previous reports [14,38], taking animal welfare into consideration.

### 4.2. Preparation of CSE

CSE was prepared as described previously [48]. Briefly, a reference cigarette (1R6F) was smoked under the Health Canada Intense smoking regime (55 mL puff volume, 30 s puff interval, 2 s puff duration, bell-shaped puff profile, and 100% blocked filter ventilation holes), using an analytical vaping machine (LM5E; Körber Technologies Instruments GmbH, Hamburg, Germany). The particulate phase of the mainstream smoke was collected on a 44 mm Cambridge filter pad (Körber Technologies Instruments GmbH). The gas phase from ten cigarettes (eight puffs from one cigarette per trial) was bubbled into 5 mL of ice-cold PBS to prepare CSE from a total of 80 puffs. This solution was defined as 100% CSE.

### 4.3. Intratracheal CSE-Induced COPD Model

A mouse model of COPD was induced by intratracheal instillation of 100 µL of CSE (100% concentration) on days 1, 8, and 15 [49,50]. Control mice received 100 µL of PBS at the same time points. On day 22, lung physiology measurements, bronchoalveolar lavage (BAL), and blood sampling were performed. The mice were then sacrificed for histological lung evaluation (Figure 1A). Mice were randomly assigned to groups with balanced allocation to avoid evaluating specific populations on the same day. Drug administration, respiratory function measurements, and histological evaluations were conducted accordingly.

### 4.4. TM5441 Administration

The PAI-1 inhibitor TM5441 was synthesized by Dr. Toshio Miyata, and its characteristics and specificity have been described previously [37]. TM5441 was dissolved in 0.5% methylcellulose (Fujifilm Wako, Osaka, Japan) [37] and administered daily via oral gavage at a dose of 20 mg/kg using a sonde from day 1 to day 22. This dose was based on previous reports [45,51]. Control mice received 0.5% methylcellulose at the same time points.

### 4.5. Histopathological Analysis

The left lung was fixed in 4% paraformaldehyde, embedded in paraffin, sectioned, and stained with HE. Lung emphysema and alveolar wall destruction were quantified using the MLI and DI, as described previously [17,52,53]. Briefly, MLI was measured by dividing the length of a line drawn across the lung section by the total number of intercepts counted along this line, representing the average alveolar diameter. DI was calculated by dividing the number of destructively altered alveoli by the total number of alveoli. MLI and DI were analyzed using ImageJ (version 1.54g; National Institutes of Health, Bethesda, MD, USA).

### 4.6. Lung Physiology and BAL

Airway resistance and dynamic compliance were measured using a Fine Pointe RC system (Buxco Electronics, Inc., Wilmington, NC, USA), as described previously [54,55]. After lung physiology measurements, BAL was performed three times using 1 mL of PBS per lavage. BAL cells were analyzed using flow cytometry, as described previously [55]. Briefly, BALF cells were blocked with Fc block (anti-CD16/CD32, clone 93; BioLegend, San Diego, CA, USA, 101302; 1:200 dilution) and stained with the following fluorochrome-conjugated antibodies: fluorescein isothiocyanate (FITC)-labeled anti-CD45 (clone 30-F11, 103108, BioLegend, 1:400 dilution), phycoerythrin (PE)-labeled anti-Siglec-F (clone E50-2440, 562068, BD Biosciences, San Jose, CA, USA, 1:500 dilution), allophycocyanin (APC)-Cy7-labeled anti-CD11c (clone N418, 117324, BioLegend, 1:400 dilution), APC-labeled anti-Ly-6G/Ly-6C (Gr-1) (clone RB6-8C5, 108412, BioLegend, 1:600 dilution), and Pacific Blue-labeled anti-CD11b (clone M1/70, 101224, BioLegend, 1:500 dilution). Flow cytometry was performed using a Gallios flow cytometer (Beckman Coulter, Brea, CA, USA), and data were analyzed using FlowJo software (version 10.8.1, FlowJo, LLC, Ashland, OR, USA). Cells were gated in the following order: debris was excluded based on forward and side scatter (FSC-A and SSC-A). As no viability dye was used, debris and presumed dead cells were excluded based on their characteristic FSC-A and SSC-A profiles. The main population was then defined based on size and granularity. Singlets were selected by FSC-H vs. FSC-A, and CD45^+^ leukocytes were subsequently identified. Immune cell subsets were defined as follows: macrophages were identified as CD45^+^ Siglec-F^+^ CD11c^+^ cells, and neutrophils as CD45^+^, Gr-1^+^, CD11b^+^ cells.

### 4.7. RT-PCR

The right lower lung was collected and stored in RNA Protect Tissue Reagent (76104, Qiagen, Hilden, Germany). RNA was isolated using an RNeasy Plus Mini Kit (74134, Qiagen) according to the manufacturer’s protocol. Complementary DNA (cDNA) was synthesized from the RNA using ReverTraAce qPCR RT Master Mix (FSQ-301, TOYOBO, Osaka, Japan) according to the manufacturer’s protocol. Quantitative real-time PCR was performed using THUNDERBIRD SYBR qPCR Mix (QPS-201, TOYOBO). Total RNA was reverse transcribed at 37 °C for 15 min, followed by denaturation at 95 °C for 5 min and 40 cycles of amplification (10 s at 95 °C and 30 s at 60 °C). The specific primer pairs used for qRT-PCR were as follows: PAI-1, forward, 5′-AGG ATC GAG GTA AAC GAG AGC-3′; PAI-1, reverse, 5′-GCG GGC TGA GAT GAC AAA-3′; NE, forward, 5′-GTG GTG ACT AAC ATG TGC CG-3′; NE, reverse, 5′-AAT CCA GAT CCA CAG CCT CC-3′; β-actin, forward, 5′-AAG GCC AAC CGT GAA AAG AT-3′; β-actin, reverse, 5′-GTG GTA CGA CCA GAG GCA TAC-3′. Gene expression levels were analyzed using the ΔΔ Ct method and normalized to β-actin as the housekeeping gene.

### 4.8. ELISA

Plasma PAI-1 activity (IMSPAI1KTA, Innovative Research, Novi, MI, USA) and NE levels in lung tissue (MELA20, R&D Systems, Minneapolis, MN, USA) were measured using ELISA kits, following the manufacturers’ recommended protocols.

### 4.9. Coagulation Times

Prothrombin time, activated partial thromboplastin time, and plasma fibrinogen levels were measured by the Research Institute for Animal Science in Biochemistry and Toxicology (Sagamihara, Japan) using the light-scattering method (CA-650, Sysmex Corp., Hyogo, Japan).

### 4.10. Statistical Analysis

Results were compared between groups using Student’s *t*-test and one-way ANOVA followed by Tukey’s multiple comparisons test. Data are presented as mean ± SEM. A *p*-value < 0.05 was considered statistically significant. All analyses were performed using EZR on R Commander version 1.61 (Saitama Medical Center, Jichi Medical University, Saitama, Japan) [56] and GraphPad Prism 8 (GraphPad Software, Boston, MA, USA).

## 5. Conclusions

In conclusion, this study demonstrated that PAI-1 plays a critical role in CSE-induced pulmonary inflammation and emphysema. Treatment with the PAI-1 inhibitor TM5441 can effectively ameliorate the pathological and functional changes in a murine model of COPD, without affecting coagulation times. These findings highlight PAI-1 inhibition as a promising therapeutic strategy for cigarette smoke-induced COPD.

## Figures and Tables

**Figure 1 ijms-26-07086-f001:**
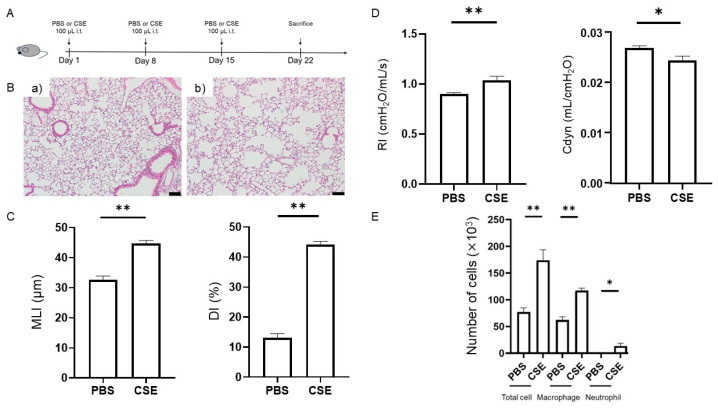
Assessment of COPD model mice following intratracheal administration of CSE. (**A**) Experimental protocol. (**B**) Representative images of HE-stained lung tissue (magnification ×100, scale bar = 100 μm): (**a**) intratracheal administration of PBS, (**b**) intratracheal administration of CSE. (**C**) MLI and DI in mice following intratracheal administration of PBS or CSE (*n* = 5 per group). (**D**) RI and Cdyn values in mice following intratracheal administration of PBS or CSE (*n* = 8–10 per group). (**E**) Cell composition in BALF (*n* = 7–8 per group). Data presented as mean ± standard error (SEM). Results were compared between groups using Student’s *t*-test. * *p* < 0.05, ** *p* < 0.01. PBS, phosphate-buffered saline; CSE, cigarette smoke extract; COPD, chronic obstructive pulmonary disease; HE, hematoxylin and eosin; MLI, mean linear intercept; DI, destructive index; RI, airway resistance; Cdyn, dynamic compliance; BALF, bronchoalveolar lavage fluid; i.t., intratracheal.

**Figure 2 ijms-26-07086-f002:**
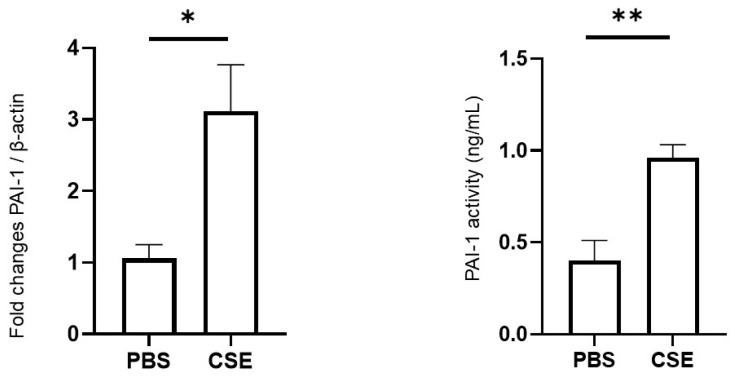
PAI-1 expression in lung tissue was assessed by RT-PCR, and plasma PAI-1 activity was measured by ELISA. Relative mRNA levels were normalized to β-actin, and fold changes are shown. Data presented as mean ± SEM (*n* = 5 per group). Results were compared between groups using Student’s *t*-test. * *p* < 0.05, ** *p* < 0.01. PBS, phosphate-buffered saline; CSE, cigarette smoke extract; PAI-1, plasminogen activator inhibitor-1.

**Figure 3 ijms-26-07086-f003:**
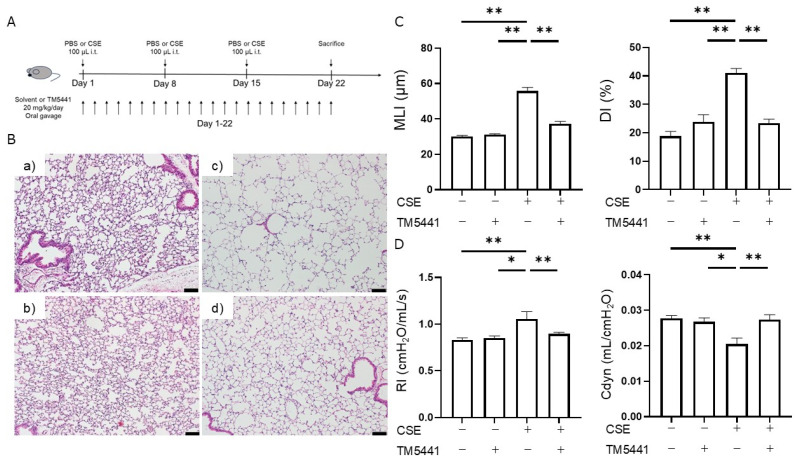
Efficacy of TM5441 in CSE-induced COPD model mice. (**A**) Experimental protocol. (**B**) Representative images of HE-stained lung tissue (magnification ×100, scale bar = 100 μm): (**a**) intratracheal administration of PBS without TM5441 treatment, (**b**) intratracheal administration of PBS with TM5441 treatment, (**c**) intratracheal administration of CSE without TM5441 treatment, (**d**) intratracheal administration of CSE with TM5441 treatment. (**C**) MLI and DI in mice following intratracheal administration of PBS or CSE and oral gavage of vehicle or TM5441 (*n* = 5 per group). (**D**) RI and Cdyn values in mice following intratracheal administration of PBS or CSE and oral gavage of vehicle or TM5441 (*n* = 5–8 per group). Data presented as mean ± SEM. Results were compared between groups using one-way ANOVA followed by Tukey’s multiple comparisons test. * *p* < 0.05, ** *p* < 0.01. PBS, phosphate-buffered saline; CSE, cigarette smoke extract; COPD, chronic obstructive pulmonary disease; MLI, mean linear intercept; DI, destructive index; RI, airway resistance; Cdyn, dynamic compliance.

**Figure 4 ijms-26-07086-f004:**
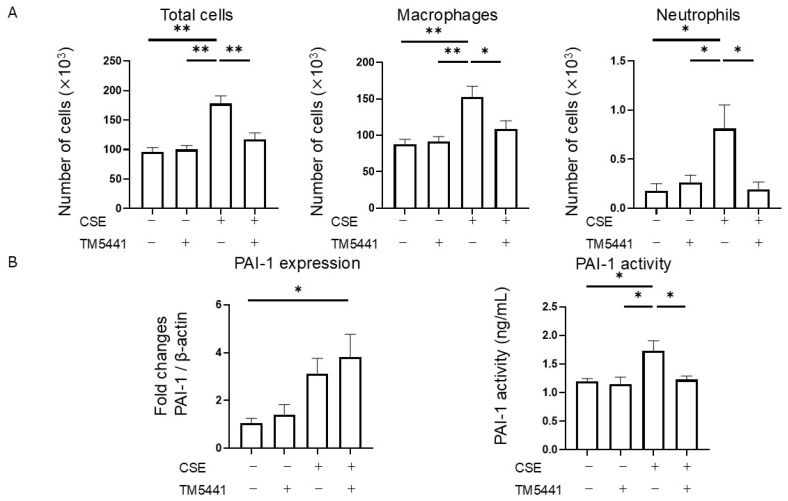
Efficacy of TM5441 on airway inflammation and PAI-1 expression in a CSE-induced mouse model of COPD. (**A**) Cell composition in BALF (*n* = 5 per group). (**B**) PAI-1 expression in lung tissue (*n* = 5 per group) was assessed by RT-PCR and PAI-1 activity in plasma (*n* = 4 per group) was measured by ELISA. Data presented as mean ± SEM. Results were compared between groups using one-way ANOVA followed by Tukey’s multiple comparisons test. * *p* < 0.05, ** *p* < 0.01. CSE, cigarette smoke extract; PAI-1, plasminogen activator inhibitor-1; BALF, bronchoalveolar lavage fluid.

**Figure 5 ijms-26-07086-f005:**
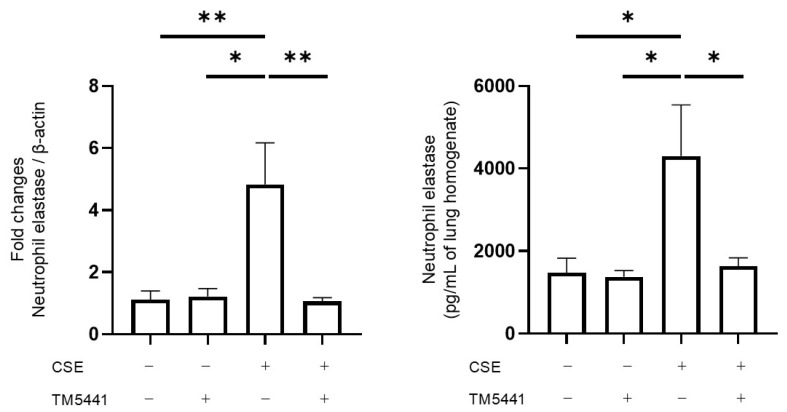
Neutrophil elastase expression in lung tissue was assessed by RT-PCR and protein levels were measured by ELISA following intratracheal administration of PBS or CSE, with or without TM5441 treatment. Relative mRNA levels were normalized to β-actin, and fold changes are shown. Data presented as mean ± SEM (*n* = 5–7 per group). Results were compared between groups using one-way ANOVA followed by Tukey’s multiple comparisons test. * *p* < 0.05, ** *p* < 0.01. PBS, phosphate-buffered saline; CSE, cigarette smoke extract; CSE, cigarette smoke extract.

**Figure 6 ijms-26-07086-f006:**
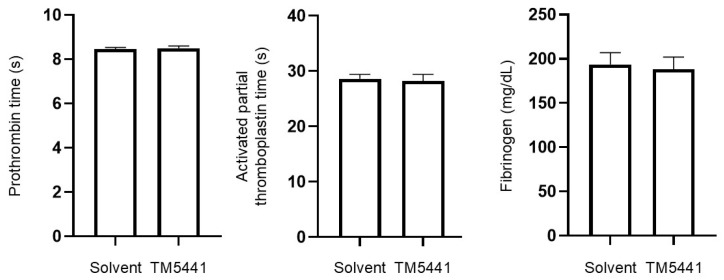
Assessment of coagulation function in plasma following intratracheal administration of cigarette smoke extract (CSE) with or without TM5441 treatment. Data presented as mean ± SEM (*n* = 4–5 per group). Results were compared between groups using Student’s *t*-test.

## Data Availability

The datasets used and analyzed during the current study are available from the corresponding author.

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
