# Peer review of "PAI-1 Inhibitor TM5441 Attenuates Emphysema and Airway Inflammation in a Murine Model of Chronic Obstructive Pulmonary Disease"

_ijms, 2025, doi:10.3390/ijms26157086_

Round 1

Reviewer 1 Report

Comments and Suggestions for Authors

This manuscript presents a well-designed and timely study investigating the therapeutic potential of the PAI-1 inhibitor TM5441 in a cigarette smoke extract-induced murine model of COPD. The results are presented well; however, several important methodological and interpretive issues need to be addressed before the manuscript can be accepted for publication:

  1. Only male mice were used in the study. Given sex-specific differences in COPD phenotypes, please justify the exclusion of female mice.
  2. Clarify whether group allocation, treatment administration, lung-function testing, and histological scoring were performed under randomized and/or blinded conditions.
  3. Discuss potential off-target effects of TM5441.
  4. Incorporating tail-bleeding time would further support the safety profile of TM5441.
  5. The current description of flow cytometry (“as described previously”) is insufficient for reproducibility. Please specify: cell-surface markers used to define macrophages and neutrophils; antibody clones, fluorochromes, vendors, and working concentrations; gating strategy, with representative plots illustrating singlet, live-cell, and lineage gates. 
  6. Please discuss potential mechanisms by which TM5441 lowers NE mRNA and protein levels.
  7. Cell counts are presented as "178 ± 13 × 103," which is ambiguous. Please use full numbers with separators "178,000 ± 13,000" or scientific notation "178  × 103 ± 13 × 103."
  8. The manuscript would benefit from discussing the strengths and limitations of the study. 

Author Response

Reviewer 1

We sincerely appreciate the reviewer’s valuable and insightful comments.

1. Only male mice were used in the study. Given sex-specific differences in COPD phenotypes, please justify the exclusion of female mice.

Response: Thank you for your comments. Sex differences in blood coagulation have been reported in mice [41,42]. Since this study involves a drug targeting the coagulation and fibrinolysis system, we standardized the sex of the animals used. Given that the prevalence of COPD is higher in males [43,44], male mice were selected to better reflect the clinical population. However, the exclusion of female mice is a limitation of this study, as noted in the manuscript (P10L330-334).

2. Clarify whether group allocation, treatment administration, lung-function testing, and histological scoring were performed under randomized and/or blinded conditions.

Response: Thank you for your important comments. In this study, mice were randomly assigned to groups with balanced allocation to avoid evaluating specific populations on the same day. Drug administration, respiratory function measurements, and histological evaluations were conducted accordingly. This has been described in the Materials and Methods section (P10L368-371). Blinding was not applied to drug administration or other procedures.

3. Discuss potential off-target effects of TM5441.

Response: Thank you for your important comment. TM5441 treatment did not alter PAI-1 mRNA expression in lung tissue following CSE administration, but it did suppress the CSE-induced increase in plasma PAI-1 activity (Figure 4B). These findings suggest that TM5441 does not affect the upstream regulation of PAI-1 expression but rather directly inhibits PAI-1 activity. Therefore, the possibility of an off-target effect of TM5441 appears unlikely. We have included these points in the Discussion section (P9L304-309).

4. Incorporating tail-bleeding time would further support the safety profile of TM5441.

Response: We appreciate your helpful suggestion. A previous report demonstrated that administration of TM5441 at doses of 50 and 100 mg/kg did not affect bleeding time in mice [45]. Therefore, we did not assess tail bleeding time in this study. This point has been added to the manuscript as a limitation (P10L334-337).

5. The current description of flow cytometry (“as described previously”) is insufficient for reproducibility. Please specify: cell-surface markers used to define macrophages and neutrophils; antibody clones, fluorochromes, vendors, and working concentrations; gating strategy, with representative plots illustrating singlet, live-cell, and lineage gates.

Response: Thank you for your insightful comment. We agree that the original description of the flow cytometry methods lacked sufficient detail for reproducibility. In the revised manuscript, we have added a comprehensive description of the flow cytometry protocol, including the cell-surface markers used to identify macrophages (e.g., CD45+ Siglec-F+ CD11c+ cells) and neutrophils (e.g., CD45+, Gr-1+, CD11b+ cells), detailed antibody information, including clones, fluorochromes, vendors, and working concentrations; and gating strategy. Representative flow cytometry plots have been included in the supplementary figure. These additions enhance clarity and reproducibility for readers and reviewers. All changes have been incorporated into the Materials and Methods section (P11L393-406), the Results section (P3L113-114), and the supplementary material has been updated accordingly.

6. Please discuss potential mechanisms by which TM5441 lowers NE mRNA and protein levels.

Response: Thank you for your comment. A previous report demonstrated that PAI-1 knockout suppressed cigarette smoke-induced myeloperoxidase activity and the expression of chemokines (CXCL1 and CXCL2) in lung tissues [12]. Furthermore, in a model of LPS-induced acute lung injury, increased neutrophil apoptosis was observed in the lungs of PAI-1 knockout mice [40]. These findings suggest that the mechanism by which TM5441 reduces NE mRNA and protein levels in the lungs is not only a reduction in neutrophil infiltration, but also inhibition of neutrophil activation and promotion of neutrophil apoptosis. These points have been included in the Discussion section (P9L314-321).

7. Cell counts are presented as "178 ± 13 × 103," which is ambiguous. Please use full numbers with separators "178,000 ± 13,000" or scientific notation "178 × 103 ± 13 × 103."

Response: Thank you for your comment. As you pointed out, we have revised all relevant numerical values to ensure consistent use of scientific notation (e.g., 178 × 103 ± 13 × 103). These changes have been implemented throughout the manuscript accordingly (P3L114-116, P5L176-178).

8. The manuscript would benefit from discussing the strengths and limitations of the study. 

Response: Thank you for your valuable suggestion. In response, we have incorporated a section outlining the strengths and limitations of the study into the manuscript (P9L322-P10L342).

Reviewer 2 Report

Comments and Suggestions for Authors

Comments to the Authors

Overall Impression: This manuscript is well-written and addresses an important gap in COPD research by evaluating a novel therapeutic strategy. The experimental design is comprehensive – spanning histological, functional, and molecular assessments – and the data show convincingly that the PAI-1 inhibitor TM5441 can ameliorate cigarette smoke extract (CSE)-induced lung damage in mice. The findings are timely given the need for new anti-inflammatory approaches in COPD, and the inclusion of safety (coagulation) data is a notable strength. There are, however, several areas where the study’s clarity, context, and interpretation could be improved. I offer the following major and minor suggestions to strengthen the manuscript:

Major Comments

Model Design and Clinical Relevance: While the acute intratracheal CSE mouse model successfully reproduces key COPD features, its limitations should be acknowledged. This model induces lung injury over a short term, which may not fully recapitulate the gradual, chronic nature of human COPD. Moreover, TM5441 was given prophylactically (starting on day 1), essentially testing preventive efficacy rather than treatment of established disease. These factors could limit the direct translational relevance. Please discuss these limitations in the Discussion. For instance, you might note that future studies using a chronic smoke exposure model or therapeutic dosing (initiating TM5441 after COPD features develop) would be valuable to confirm that PAI-1 inhibition can reverse or halt progression of existing COPD pathology. A brief acknowledgement of this point will provide a more balanced view of the findings.

Mechanistic Insight – Inflammation Pathways: The data show that TM5441 reduced neutrophilic inflammation (lower BAL neutrophil counts and lung neutrophil elastase [NE] levels), suggesting PAI-1 drives COPD pathogenesis at least in part by promoting neutrophil recruitment/activation. However, the mechanistic link between PAI-1 activity and neutrophil influx could be elaborated further. The Discussion touches on PAI-1’s role in neutrophil chemotaxis and efferocytosis, but the manuscript would benefit from deeper analysis or additional data on inflammatory mediators:

Suggestion: If available, consider reporting changes in key neutrophil chemoattractants or cytokines (for example, CXCL1/CXCL2 or TNF-α) in the lungs or BAL fluid of CSE vs. CSE+TM5441 groups. This would strengthen the conclusion that PAI-1 inhibition blunts upstream signals driving neutrophil recruitment. Even if such data are not available, expanding the discussion to hypothesize how PAI-1 promotes neutrophilic inflammation (e.g., via upregulation of C-X-C chemokines as noted in prior studies) would provide more mechanistic context. Additionally, it would be useful to clarify whether the reduction in NE mRNA/protein is simply a downstream consequence of fewer neutrophils in the lungs, or if PAI-1 might directly influence NE expression/release from neutrophils. Currently, the results imply that TM5441’s effect on NE is likely mediated by reduced neutrophil infiltration, but a brief comment on this point would be helpful for readers.

Data Presentation and Clarity: A few results and terms should be clarified to avoid confusion:

BAL Neutrophil Counts: In Section 2.3, CSE exposure increased BAL neutrophils from ~0.4×10^3 to 13.3×10^3 cells. However, in Section 2.7 (with TM5441 treatment), the CSE group’s neutrophil count is reported as 0.8×10^3 (reduced to 0.2×10^3 with TM5441). This is a much lower absolute neutrophil number than the 13.3×10^3 seen earlier for CSE alone. Please clarify this apparent discrepancy. If the data in Section 2.7 come from a separate experiment or time point, explain why the baseline CSE neutrophil count differs (e.g., differences in animal cohorts or techniques). Ensuring consistency in units and notation (e.g. expressing all cell counts in the same units of 10^3 cells) will help the reader compare these results.

Abbreviations (RI and Cdyn): The terms RI and Cdyn should be defined upon first use. Readers may not immediately recognize RI as airway resistance and Cdyn as dynamic compliance. Although these are defined in the Figure 3 caption, please also define them in the main text (Section 2.2) or include them in the abbreviations list for clarity.

NE Protein Units: When reporting neutrophil elastase protein levels (Section 2.9), clarify the units and basis – for example, “NE protein levels (pg/mL of lung homogenate)”. Stating that these values were derived from lung tissue homogenate ELISA (as per Methods) will prevent any confusion about the meaning of “pg/mL” in this context.

Statistical detail: It’s noted that most group comparisons yielded very small p-values, indicating robust differences. One exception was the BAL neutrophil increase with CSE (p≈0.022, Section 2.3), which suggests some variability. It would be reassuring to mention the sample sizes (n) for each comparison in the Results text (as you have done in figure legends) so that readers appreciate the statistical power. For instance, explicitly stating “(n=5 per group)” for the BAL cell counts in the text would align with the figure and methods and emphasize that these findings are supported by appropriate replicates.

Contextualization and Novelty: The manuscript could better highlight how these findings advance current knowledge and how TM5441 compares to other potential COPD therapies:

Novelty: Prior studies have implicated PAI-1 in COPD (e.g., genetic PAI-1 deficiency protected mice from smoke-induced inflammation), and PAI-1 inhibitors have shown efficacy in other lung disease models. Your study is novel in demonstrating the efficacy of a PAI-1 inhibitor in a cigarette smoke-induced COPD model. Please emphasize this point in the Introduction or Discussion – for instance, by explicitly stating that this is the first evaluation of a PAI-1 inhibitor in a COPD context, filling an important gap in preclinical research. Making the novelty clear will strengthen the impact of the work.

Comparison to other anti-inflammatories: It would enrich the Discussion to briefly compare targeting PAI-1 with other anti-inflammatory or protease-targeting strategies in COPD. For example, neutrophil elastase inhibitors have been tested clinically in COPD but with limited success (AZD9668, an oral NE inhibitor, showed no clinical benefit in COPD trials). Similarly, broad anti-inflammatory approaches (e.g., high-dose inhaled steroids or PDE4 inhibitors like roflumilast) yield only modest reductions in COPD exacerbations. By highlighting these points, you can position PAI-1 inhibition as a potentially promising alternative that targets an upstream driver of both inflammation and tissue destruction. This perspective would underscore the translational relevance of your findings.

Future directions: Consider adding 1–2 sentences in the Discussion or Conclusions about possible next steps. For instance, you might suggest investigating TM5441 in combination with existing therapies, or evaluating long-term effects on small-airway remodeling and fibrosis (since PAI-1 is profibrotic, one wonders if inhibition might also reduce any fibrotic/structural remodeling in chronic disease). Acknowledging such aspects will show awareness of the broader context and limitations of the current study, without detracting from your results.

Minor Comments

Section 2.2 Heading: The heading “2.2. CSE-induced lung functional …” appears to be incomplete or oddly phrased. It likely should read something like “CSE-induced lung functional impairment” or “CSE-induced changes in lung function.” Please revise this section title for clarity.

Section 2.8 Heading: “plasmas PAI-1 activity” should be corrected to “plasma PAI-1 activity.”

Affiliation Capitalization: In the author affiliations, “Department of pharmaceutical Sciences” should be “Department of Pharmaceutical Sciences” (capitalize “Pharmaceutical” for consistency with other department names).

Abbreviation List: Please add RI (airway resistance) and Cdyn (dynamic compliance) to the abbreviations list, or ensure they are defined in the text, as noted above. This will help readers unfamiliar with these terms.

Formatting of Units and Symbols: Check for consistency in reporting units and symbols. For example, in several places a space is included before the percent sign (e.g., “13.2 ± 1.4 %”), which is not standard; it should be “13.2 ± 1.4%” without a space. Similarly, ensure there is a space between numbers and units (e.g., “100 μm” rather than “100μm” if any appear). These are minor typesetting issues, but attention to journal style will improve readability.

Figure References and Labels: All figure panels should be clearly labeled and referenced in the text. In Section 2.1, for instance, you refer to Figure 1A, 1B, 1C, etc. Make sure each panel is described. (It appears Figure 1A shows the experimental schematic, 1B the histology images, 1C the MLI/DI quantification, and 1D the lung mechanics data – this is clear enough, but double-check that the lettering in the figure images matches the citations in text.) Also, when citing multiple figure panels together, the journal might prefer the format “Figure 1B,C” without a space. These are minor editorial details that can be adjusted during revision.

Reference Section: In the reference list of the submitted PDF, there is a heading “Uncategorized References.” This seems to be a formatting error. Ensure that the final reference list is properly formatted according to IJMS style (all references should be numbered and ordered as they appear in the text). It looks like the references themselves are correctly listed; removing the word “Uncategorized” will clean up the appearance.

Typographical Errors: The manuscript is generally very well edited (kudos on the clear writing!). Just fix a few typos such as extra hyphenation due to line breaks (e.g., “pathophysi-ological” in line 564–565 should read “pathophysiological” once the line break is removed). These are minor issues likely introduced by the formatting and will be resolved in proofreading.

By addressing these comments, the authors can further improve the clarity and impact of this work. The study’s results are exciting and suggest that PAI-1 inhibition could be a promising therapeutic avenue in COPD. I congratulate the authors on a strong set of experiments and encourage them to incorporate the revisions above. With these adjustments, the manuscript will be in excellent shape for publication.

Comments on the Quality of English Language

Overall, the manuscript is written in clear and professional English, but a careful copy-edit would enhance readability and consistency. Minor issues include occasional typographical errors (e.g., “plasmas PAI-1 activity,” “pathophysi-ological”), redundant wording (“significantly … significantly”), and inconsistent spacing around symbols and units (e.g., “13.2 ± 1.4 %” should be “13.2 ± 1.4%”). Some section headings require rephrasing for fluency (e.g., “CSE-induced lung functional” → “CSE-induced changes in lung function”), and key abbreviations such as RI and Cdyn should be defined on first use and/or added to the abbreviation list. Ensuring uniform terminology (distinguishing PAI-1 “activity” from “expression”) and standardizing figure captions will further improve clarity. A final language polish—either by the authors or a professional editing service—will address these minor issues efficiently.

Author Response

Reviewer 2

We sincerely appreciate the reviewer’s valuable and insightful comments. 

Major Comments

Model Design and Clinical Relevance: While the acute intratracheal CSE mouse model successfully reproduces key COPD features, its limitations should be acknowledged. This model induces lung injury over a short term, which may not fully recapitulate the gradual, chronic nature of human COPD. Moreover, TM5441 was given prophylactically (starting on day 1), essentially testing preventive efficacy rather than treatment of established disease. These factors could limit the direct translational relevance. Please discuss these limitations in the Discussion. For instance, you might note that future studies using a chronic smoke exposure model or therapeutic dosing (initiating TM5441 after COPD features develop) would be valuable to confirm that PAI-1 inhibition can reverse or halt the progression of existing COPD pathology. A brief acknowledgement of this point will provide a more balanced view of the findings.

Response: We are grateful for your constructive comments. As you pointed out, this experimental model does not fully replicate COPD associated with chronic exposure to cigarette smoke, and the timing of TM5441 administration is a limitation with respect to evaluating its therapeutic effects. Accordingly, we have acknowledged this as a limitation in the manuscript (P10L338-342). “This experimental model does not fully reflect COPD associated with chronic cigarette exposure. Furthermore, the timing of TM5441 administration limits the ability to evaluate its preventive effects. In future studies, we plan to use an experimental design in which drug administration begins after disease onset, in order to assess whether PAI-1 inhibition can reverse or halt the progression of existing COPD pathology. Finally, given that PAI-1 has profibrotic effects and is associated with airway fibrosis, and that small-airway remodeling is a key pathological feature of COPD, we plan to investigate whether TM5441 can prevent long-term small-airway remodeling and fibrosis in future studies.” In future studies, we intend to consider an experimental design in which drug administration begins after disease onset (P10L340-341).

Mechanistic Insight – Inflammation Pathways: The data show that TM5441 reduced neutrophilic inflammation (lower BAL neutrophil counts and lung neutrophil elastase [NE] levels), suggesting PAI-1 drives COPD pathogenesis at least in part by promoting neutrophil recruitment/activation. However, the mechanistic link between PAI-1 activity and neutrophil influx could be elaborated further. The Discussion touches on PAI-1’s role in neutrophil chemotaxis and efferocytosis, but the manuscript would benefit from deeper analysis or additional data on inflammatory mediators:

Suggestion: If available, consider reporting changes in key neutrophil chemoattractants or cytokines (for example, CXCL1/CXCL2 or TNF-α) in the lungs or BAL fluid of CSE vs. CSE+TM5441 groups. This would strengthen the conclusion that PAI-1 inhibition blunts upstream signals driving neutrophil recruitment. Even if such data are not available, expanding the discussion to hypothesize how PAI-1 promotes neutrophilic inflammation (e.g., via upregulation of C-X-C chemokines as noted in prior studies) would provide more mechanistic context. Additionally, it would be useful to clarify whether the reduction in NE mRNA/protein is simply a downstream consequence of fewer neutrophils in the lungs or if PAI-1 might directly influence NE expression/release from neutrophils. Currently, the results imply that TM5441’s effect on NE is likely mediated by reduced neutrophil infiltration, but a brief comment on this point would be helpful for readers.

Response: Thank you for your valuable suggestion. A previous report demonstrated that PAI-1 knockout suppressed cigarette smoke-induced myeloperoxidase activity and the expression of chemokines (CXCL1 and CXCL2) in lung tissues [12]. Furthermore, in a model of LPS-induced acute lung injury, increased neutrophil apoptosis was observed in the lungs of PAI-1 knockout mice [40]. These findings suggest that the mechanism by which TM5441 reduces NE mRNA and protein levels in the lungs is not only a reduction in neutrophil infiltration, but also inhibition of neutrophil activation and promotion of neutrophil apoptosis. These points have been included in the Discussion section (P9L314-321).

Data Presentation and Clarity: A few results and terms should be clarified to avoid confusion: BAL Neutrophil Counts: In Section 2.3, CSE exposure increased BAL neutrophils from ~0.4×10^3 to 13.3×10^3 cells. However, in Section 2.7 (with TM5441 treatment), the CSE group’s neutrophil count is reported as 0.8×10^3 (reduced to 0.2×10^3 with TM5441). This is a much lower absolute neutrophil number than the 13.3×10^3 seen earlier for CSE alone. Please clarify this apparent discrepancy. If the data in Section 2.7 come from a separate experiment or time point, explain why the baseline CSE neutrophil count differs (e.g., differences in animal cohorts or techniques). Ensuring consistency in units and notation (e.g., expressing all cell counts in the same units of 10^3 cells) will help the reader compare these results.

Response: Thank you for your insightful comment regarding the apparent discrepancy in BAL neutrophil counts between sections 2.3 and 2.7. As you pointed out, the cell counts were presented with consistent units and notation (× 103). The difference in absolute neutrophil counts observed between the two sections is primarily due to inter-experimental variability. Specifically, the data presented in section 2.7 were obtained from a separate set of experiments conducted at a different time from those in section 2.3. Although the mice used in both experiments were of the same strain, sex, and age, the experiments were performed at different times using independently housed cohorts. Even under standardized conditions, neutrophil counts in BALF were lower than those of macrophages and varied among individual mice following CSE administration.

Abbreviations (RI and Cdyn): The terms RI and Cdyn should be defined upon first use. Readers may not immediately recognize RI as airway resistance and Cdyn as dynamic compliance. Although these are defined in the Figure 3 caption, please also define them in the main text (Section 2.2) or include them in the abbreviations list for clarity.

Response: Thank you for your helpful comment. In response, we have added definitions for both terms at their first mention in Section 2.2 (P3L105-106). Additionally, we have included both abbreviations in the Abbreviations list at the end of the manuscript.

NE Protein Units: When reporting neutrophil elastase protein levels (Section 2.9), clarify the units and basis – for example, “NE protein levels (pg/mL of lung homogenate)”. Stating that these values were derived from lung tissue homogenate ELISA (as per Methods) will prevent any confusion about the meaning of “pg/mL” in this context.

Response: Thank you for your suggestion. In the revised manuscript, we now explicitly state that NE levels were measured in lung tissue homogenates using ELISA and expressed as pg/mL of lung homogenate accordingly.

Statistical detail: It’s noted that most group comparisons yielded very small p-values, indicating robust differences. One exception was the BAL neutrophil increase with CSE (p≈0.022, Section 2.3), which suggests some variability. It would be reassuring to mention the sample sizes (n) for each comparison in the Results text (as you have done in figure legends) so that readers appreciate the statistical power. For instance, explicitly stating “(n=5 per group)” for the BAL cell counts in the text would align with the figure and methods and emphasize that these findings are supported by appropriate replicates.

Response: Thank you for your comment. We have added the sample sizes (e.g., n = 5 per group) to the relevant sections of the results to clarify the statistical context and ensure consistency.

Contextualization and Novelty: The manuscript could better highlight how these findings advance current knowledge and how TM5441 compares to other potential COPD therapies:

Novelty: Prior studies have implicated PAI-1 in COPD (e.g., genetic PAI-1 deficiency protected mice from smoke-induced inflammation), and PAI-1 inhibitors have shown efficacy in other lung disease models. Your study is novel in demonstrating the efficacy of a PAI-1 inhibitor in a cigarette smoke-induced COPD model. Please emphasize this point in the Introduction or Discussion – for instance, by explicitly stating that this is the first evaluation of a PAI-1 inhibitor in a COPD context, filling an important gap in preclinical research. Making the novelty clear will strengthen the impact of the work.

Response: Thank you for your helpful advice. We have incorporated these points as strengths in the Discussion section (P9L322-P10L329). To our knowledge, this is the first report to demonstrate the efficacy of a PAI-1 inhibitor in a cigarette smoke-induced COPD model. Targeting PAI-1 represents a novel anti-inflammatory approach for COPD.

Comparison to other anti-inflammatories: It would enrich the Discussion to briefly compare targeting PAI-1 with other anti-inflammatory or protease-targeting strategies in COPD. For example, neutrophil elastase inhibitors have been tested clinically in COPD but with limited success (AZD9668, an oral NE inhibitor, showed no clinical benefit in COPD trials). Similarly, broad anti-inflammatory approaches (e.g., high-dose inhaled steroids or PDE4 inhibitors like roflumilast) yield only modest reductions in COPD exacerbations. By highlighting these points, you can position PAI-1 inhibition as a potentially promising alternative that targets an upstream driver of both inflammation and tissue destruction. This perspective would underscore the translational relevance of your findings.

Response: Thank you for your valuable suggestion. We have added the following sentences to the discussion section (P8L238-245). Several anti-inflammatory drugs have been considered as candidates for COPD treatment. However, the efficacy of broad anti-inflammatory approaches remains limited, with only modest reductions in COPD exacerbations. High-dose inhaled corticosteroids carry an increased risk of pneumonia [20,21], and phosphodiesterase-4 inhibitors are associated with gastrointestinal side effects [22]. The oral neutrophil elastase inhibitor AZD9668 showed no clinical benefit in patients with COPD [23], and the development of neutrophil elastase inhibitors has not progressed further. Furthermore, PAI-1 is associated with common comorbidities of COPD such as atherosclerosis and metabolic syndrome, and PAI-1 inhibition is considered a promising therapeutic approach in clinical-practice. In this study, TM5441 prevented the pathophysiological condition of COPD, and PAI-1 inhibition may represent a promising alternative therapeutic strategy that targets an upstream driver of both inflammation and tissue destruction.

Future directions: Consider adding 1–2 sentences in the Discussion or Conclusions about possible next steps. For instance, you might suggest investigating TM5441 in combination with existing therapies, or evaluating long-term effects on small-airway remodeling and fibrosis (since PAI-1 is profibrotic, one wonders if inhibition might also reduce any fibrotic/structural remodeling in chronic disease). Acknowledging such aspects will show awareness of the broader context and limitations of the current study, without detracting from your results.

Response: Thank you for your important comments. We have added to the Discussion section indicating our plan to evaluate the long-term effects of TM5441 on small-airway remodeling and fibrosis (P10L342-346). Finally, given that PAI-1 has profibrotic effects [46] and is associated with airway fibrosis, and that small-airway remodeling is a key pathological feature of COPD [47], we plan to investigate whether TM5441 can prevent long-term small-airway remodeling and fibrosis in future studies.

Minor Comments

Section 2.2 Heading: The heading “2.2. CSE-induced lung functional …” appears to be incomplete or oddly phrased. It likely should read something like “CSE-induced lung functional impairment” or “CSE-induced changes in lung function.” Please revise this section title for clarity.

Response: Thank you for pointing this out. We have corrected the heading of section2.2 to CSE-induced lung functional impairment.

Section 2.8 Heading: “plasmas PAI-1 activity” should be corrected to “plasma PAI-1 activity.”

Response: Thank you for your suggestion. We have made corrections accordingly.

Affiliation Capitalization: In the author affiliations, “Department of pharmaceutical Sciences” should be “Department of Pharmaceutical Sciences” (capitalize “Pharmaceutical” for consistency with other department names).

Response: Thank you for your comment. We have revised the text accordingly.

Abbreviation List: Please add RI (airway resistance) and Cdyn (dynamic compliance) to the abbreviations list, or ensure they are defined in the text, as noted above. This will help readers unfamiliar with these terms.

Response: Thank you for your helpful comment. We have added definitions for both terms at their first mention. Additionally, we have included both abbreviations in the Abbreviations list at the end of the manuscript.

Formatting of Units and Symbols: Check for consistency in reporting units and symbols. For example, in several places a space is included before the percent sign (e.g., “13.2 ± 1.4 %”), which is not standard; it should be “13.2 ± 1.4%” without a space. Similarly, ensure there is a space between numbers and units (e.g., “100 μm” rather than “100μm” if any appear). These are minor typesetting issues, but attention to journal style will improve readability.

Response: Thank you for your careful observation. We have reviewed the manuscript to ensure consistency in the formatting of units and symbols. Specifically, we have removed unnecessary spaces before percentage signs (e.g., corrected “13.2 ± 1.4 %” to “13.2 ± 1.4%”) and verified that appropriate spacing is included between numbers and units (e.g., “100 μm” instead of “100μm”).

Figure References and Labels: All figure panels should be clearly labeled and referenced in the text. In Section 2.1, for instance, you refer to Figure 1A, 1B, 1C, etc. Make sure each panel is described. (It appears Figure 1A shows the experimental schematic, 1B the histology images, 1C the MLI/DI quantification, and 1D the lung mechanics data – this is clear enough, but double-check that the lettering in the figure images matches the citations in text.) Also, when citing multiple figure panels together, the journal might prefer the format “Figure 1B,C” without a space. These are minor editorial details that can be adjusted during revision.

Response: Thank you for your comments. We have reviewed all figure panels to ensure that they are clearly labeled and consistently referenced in the text. We have also verified that the lettering in the figures matches the in-text citations, and adjusted the formatting of multiple panel references in accordance with journal’s style.

Reference Section: In the reference list of the submitted PDF, there is a heading “Uncategorized References.” This seems to be a formatting error. Ensure that the final reference list is properly formatted according to IJMS style (all references should be numbered and ordered as they appear in the text). It looks like the references themselves are correctly listed; removing the word “Uncategorized” will clean up the appearance.

Typographical Errors: The manuscript is generally very well edited (kudos on the clear writing!). Just fix a few typos such as extra hyphenation due to line breaks (e.g., “pathophysi-ological” in line 564–565 should read “pathophysiological” once the line break is removed). These are minor issues likely introduced by the formatting and will be resolved in proofreading.

Response: Thank you for your kind feedback and helpful suggestions.

We have removed the heading “Uncategorized References” and confirmed that all references are properly numbered and ordered according to their appearance in the text, in line with IJMS formatting guidelines. We have also reviewed the manuscript for minor typographical issues, such as unintended hyphenation due to line breaks (e.g., pathophysiological), and corrected them accordingly.

Round 2

Reviewer 1 Report

Comments and Suggestions for Authors

The authors have addressed all previous comments. However, a few details regarding the flow cytometry implementation require further clarification:

1. Subsection 4.6. states that "debris and dead cells were excluded," yet no viability dye is listed, and no gate for live cells appears in Figure S1. Please specify in the text which viability reagent was used and include the corresponding gate in the figure.

2. There is no description of how spectral overlap was compensated. No single-colour or FMO controls are mentioned. Please clarify the compensation workflow.

3. The protocol does not indicate whether Fc block was used to reduce non-specific binding..

4. The neutrophil marker clone is RB6-8C5, not "R86-8C5."

5. According to the BioLegend datasheet, clone RB6-8C5 binds with high affinity to mouse Ly6G molecules and to a lower extent to Ly6C, which implies that the CD11b+Gr1+ gate included both neutrophils and Ly6Chi inflammatory monocytes. This should be acknowledged in the manuscript.

Author Response

Reviewer 1

We sincerely appreciate the reviewer’s valuable and insightful comments. 

  1. Subsection 4.6. states that "debris and dead cells were excluded," yet no viability dye is listed, and no gate for live cells appears in Figure S1. Please specify in the text which viability reagent was used and include the corresponding gate in the figure.

Response: Thank you for your important comment and we apologize for the confusion. In this study, we did not use a viability dye to exclude dead cells. Instead, we excluded debris and presumed dead cells based on their characteristic forward and side scatter (FSC-A and SSC-A) profiles. We have revised the text in subsection 4.6 to clarify this point (P11L414-418).

  1. There is no description of how spectral overlap was compensated. No single-colour or FMO controls are mentioned. Please clarify the compensation workflow.

Response: Thank you for your valuable comment. We have now clarified the compensation strategy and added Supplementary Figure S1A, which shows histogram overlays of fully stained samples (black) and fluorescence-minus one (FMO) controls (gray). The FMO controls were used to define the gating thresholds.

  1. The protocol does not indicate whether Fc block was used to reduce non-specific binding.

Response: Thank you for your insightful comment. In this experiment, we used an Fc block (anti-mouse CD16/CD32, clone 93; BioLegend, 101302; 1:200 dilution) to reduce non-specific binding. We have now added this information to Materials and Methods section of the revised manuscript (P11L405-406).

  1. The neutrophil marker clone is RB6-8C5, not "R86-8C5."

Response: Thank you very much for pointing out the typographical error. We have corrected the clone name from “R86-8C5” to “RB6-8C5” in the revised manuscript (P11L411).

  1. According to the BioLegend datasheet, clone RB6-8C5 binds with high affinity to mouse Ly6G molecules and to a lower extent to Ly6C, which implies that the CD11b+Gr1+ gate included both neutrophils and Ly6Chi inflammatory monocytes. This should be acknowledged in the manuscript.

Response: Thank you for your valuable comment. We agree with your point. As noted in the BioLegend datasheet, clone RB6-8C5 recognizes both Ly6G (with high affinity) and Ly6C (to a lesser extent), indicating that the CD11b+Gr-1+ population may include not only neutrophils but also Ly6Chi inflammatory monocytes. We have added a corresponding note to the Discussion section of the revised manuscript to acknowledge this limitation and clarify the interpretation of our flow cytometry data (P11L414-418).

Reviewer 2 Report

Comments and Suggestions for Authors

Dear authors, 

Overall, this is a clearly written and carefully executed pre‑clinical study that provides useful evidence supporting plasminogen‑activator–inhibitor‑1 (PAI‑1) blockade as a disease‑modifying strategy in cigarette‑smoke‑related COPD. Your in‑vivo design (three intratracheal CSE instillations followed by TM5441 gavage) and the breadth of structural, functional and molecular read‑outs are coherent and convincing. Inclusion of coagulation parameters is also welcome from a safety standpoint.

Strengths

Robust murine model: emphysema, airflow limitation and neutrophilic inflammation were reproduced and quantified (MLI, DI, RI, Cdyn, BALF counts).

Comprehensive methodology: random allocation, balanced scheduling and appropriate statistics (t‑test or one‑way ANOVA with Tukey) are reported.

Clear translational signal: TM5441 reduced air‑space enlargement, airway resistance, inflammatory cell influx and systemic PAI‑1 activity without prolonging coagulation time.

Transparent ethics and funding statements: IACUC approval and absence of conflicts are documented.

Minor suggestions (all non‑blocking)

Sex considerations – Only male mice were used; please discuss briefly how this may limit generalisability and whether female cohorts are planned for follow‑up studies.

Power justification – Add a sentence indicating how group sizes (n ≈ 5–10) were determined (e.g., pilot data or a priori power calculation).

Blinding – Clarify whether investigators analysing histology and lung‑function data were blinded to group allocation.

Data availability – Consider depositing raw datasets (BALF counts, qPCR Ct values, etc.) in a public repository rather than “upon reasonable request”.

Figures – Ensure all figure panels contain exact n values and the statistical test used; a graphical abstract could further improve accessibility.

Discussion – You already note limitations of the acute CSE exposure and concurrent treatment; please speculate briefly on how chronic smoke exposure or post‑disease‑initiation dosing might affect efficacy.

Addressing these minor points will further strengthen an already solid manuscript.

Author Response

Reviewer 2

We sincerely appreciate the reviewer’s valuable and insightful comments.

Overall, this is a clearly written and carefully executed pre‑clinical study that provides useful evidence supporting plasminogen‑activator–inhibitor‑1 (PAI‑1) blockade as a disease‑modifying strategy in cigarette‑smoke‑related COPD. Your in‑vivo design (three intratracheal CSE instillations followed by TM5441 gavage) and the breadth of structural, functional and molecular read‑outs are coherent and convincing. Inclusion of coagulation parameters is also welcome from a safety standpoint.

Strengths Robust murine model: emphysema, airflow limitation and neutrophilic inflammation were reproduced and quantified (MLI, DI, RI, Cdyn, BALF counts).

Comprehensive methodology: random allocation, balanced scheduling and appropriate statistics (t‑test or one‑way ANOVA with Tukey) are reported. Clear translational signal: TM5441 reduced air‑space enlargement, airway resistance, inflammatory cell influx and systemic PAI‑1 activity without prolonging coagulation time. Transparent ethics and funding statements: IACUC approval and absence of conflicts are documented.

Response: Thank you very much for your comments regarding the strengths.

Minor suggestions (all non‑blocking)

Sex considerations – Only male mice were used; please discuss briefly how this may limit generalisability and whether female cohorts are planned for follow‑up studies.

Response: Thank you for your comment. We added the phrase “as well as evaluate its efficacy in female mice” to the future plan section (P10L355).

Power justification – Add a sentence indicating how group sizes (n ≈ 5–10) were determined (e.g., pilot data or a priori power calculation).

Response: Thank you for your suggestion. We determined the group sizes with reference to previous reports [14,38], taking animal welfare into consideration. This has been described in the manuscript (P10L362-363).

Blinding – Clarify whether investigators analysing histology and lung‑function data were blinded to group allocation.

Response: Thank you for your comments. The investigator performed the respiratory function and tissue analyses with knowledge of the group names but without detailed information about the group assignments.

Data availability – Consider depositing raw datasets (BALF counts, qPCR Ct values, etc.) in a public repository rather than “upon reasonable request”.

Response: Thank you very much for your comment. We are considering depositing raw datasets in a public repository and have therefore removed the phrase “upon reasonable request”.

Figures – Ensure all figure panels contain exact n values and the statistical test used; a graphical abstract could further improve accessibility.

Response: Thank you very much for your comments. We have ensured that all figure panels include the exact n values and have added the statistical tests used in all figures.

Discussion – You already note limitations of the acute CSE exposure and concurrent treatment; please speculate briefly on how chronic smoke exposure or post‑disease‑initiation dosing might affect efficacy.

Response: Thank you for your comment. The therapeutic effect after disease pathogenesis has been established is expected to be more difficult to achieve than the preventive effect. To better evaluate the therapeutic potential of TM5441, it would be preferable to use a model that more accurately reflects the pathophysiology of human COPD. We added the following sentence: “It would be preferable to validate these findings using a model that more accurately reflects the pathophysiology of human COPD in order to better evaluate the therapeutic potential of TM5441 (P10L352-354)”.
